# Increased lethality in influenza and SARS-CoV-2 coinfection is prevented by influenza immunity but not SARS-CoV-2 immunity

Hagit Achdout [1,3], Einat. B. Vitner [1,3], Boaz Politi[1,3], Sharon Melamed[1,3], Yfat Yahalom-Ronen[1], Hadas Tamir[1], Noam Erez[1], Roy Avraham [1], Shay Weiss[1], Lilach Cherry[1], Erez Bar-Haim[2], Efi Makdasi[1], David Gur[2], Moshe Aftalion[2], Theodor Chitlaru[2], Yaron Vagima[2], Nir Paran [1] & Tomer Israely [1✉]

Severe acute respiratory syndrome coronavirus 2 (SARS-CoV-2) is the cause of the ongoing coronavirus disease 2019 (COVID-19) pandemic. The continued spread of SARS-CoV-2 increases the probability of influenza/SARS-CoV-2 coinfection, which may result in severe disease. In this study, we examine the disease outcome of influenza A virus (IAV) and SARS-CoV-2 coinfection in K18-hACE2 mice. Our data indicate enhance susceptibility of IAV-infected mice to developing severe disease upon coinfection with SARS-CoV-2 two days later. In contrast to nonfatal influenza and lower mortality rates due to SARS-CoV-2 alone, this coinfection results in severe morbidity and nearly complete mortality. Coinfection is associated with elevated influenza viral loads in respiratory organs. Remarkably, prior immunity to influenza, but not to SARS-CoV-2, prevents severe disease and mortality. This protection is antibody-dependent. These data experimentally support the necessity of seasonal influenza vaccination for reducing the risk of severe influenza/COVID-19 comorbidity during the COVID-19 pandemic.

[1] Departments of Infectious Diseases, Israel Institute for Biological Research, Ness-Ziona 7410001, Israel. [2] Department of Biochemistry and Molecular Genetics, Israel Institute for Biological Research, Ness-Ziona 7410001, Israel. [3] These authors contributed equally: Hagit Achdout, Einat. B. Vitner, Boaz Politi, Sharon Melamed. ✉email: tomeri@iibr.gov.il

The pandemic of coronavirus disease 2019 (COVID-19), which is caused by severe acute respiratory syndrome coronavirus 2 (SARS-CoV-2), has posed serious threats to global health and economy. The COVID-19 pandemic presents a broad spectrum of severities, ranging from asymptomatic presentation to severe pneumonia. Most of the mildly to critically affected patients show respiratory complications, including moderate to severe pneumonia, which can further progress to acute respiratory distress syndrome, sepsis, and multiple organ dysfunction in severely ill patients[1]. Most of these clinical symptoms are associated with the respiratory system, specifically the lungs, resulting in the depleted lung functionality[2]. Several risk factors for severe COVID-19 disease, such as the age, sex, and obesity have been identified[3]. However, more research is needed to better understand the involvement of these and other risk factors in severe illness or complications. Whether coinfection with other pathogens may impact disease severity is yet to be elucidated. Previous studies have reported coinfections between SARS-CoV-2 and common respiratory viruses, including rhinovirus, influenza virus, metapneumovirus, parainfluenza virus, and respiratory syncytial virus[4]. A meta-analysis of 30 studies including 3834 patients with COVID-19 revealed that overall, 7% of hospitalized patients with COVID-19 had a bacterial coinfection, while the pooled proportion of viral coinfections was 3%, with respiratory syncytial virus and influenza A virus (IAV) being the most common[5]. However, the prevalence and disease outcomes of viral coinfections in the SARS-CoV-2-positive population remain controversial[6–9].

IAV is a leading cause of respiratory infection, resulting in respiratory disease[10]. Complications involving secondary infections with other pathogens, mostly bacteria, significantly exacerbate the risk of severe disease due to IAV and other viral respiratory pathogens[11–14].

However, whether coinfection with IAV and SARS-CoV-2 presents more severe disease than a single infection is not clear.

In this study, we delineate the interplay between IAV and SARS-CoV-2 infections. For this purpose, we employ transgenic mice expressing human angiotensin-converting enzyme 2 (hACE2) under control of the human cytokeratin 18 promoter (K18-hACE2 mice) to establish a SARS-CoV-2-susceptible murine model[15]. We show that IAV-infected mice have enhanced susceptibility to developing severe disease upon coinfection with SARS-CoV-2 two days post influenza infection (dpIi). In contrast to nonfatal influenza disease and lower rates of mortality due to SARS-CoV-2 alone, this coinfection is associated with elevated influenza viral loads in respiratory organs, while SARS-CoV-2 viral load was reduced following coinfection. Remarkably, prior immunity to influenza, but not to SARS-CoV-2, prevents severe disease and mortality. Finally, we show that this protection is antibody dependent.

## Results and discussion

**Severe mortality in K18-hACE2 mice coinfected with SARS-CoV-2 and IAV**. To delineate the interplay between IAV and SARS-CoV-2 infections, we used K18-hACE2 mice to establish a SARS-CoV-2-susceptible murine model[15]. Non/partially lethal viral doses were chosen for each virus to allow us to distinguish more severe outcomes of coinfection. The doses were determined based on challenge experiments in which K18-hACE2 mice were infected with different doses of mouse-adapted IAV (A/Puerto Rico/8/1934 H1N1 (PR8)) or SARS-CoV-2 (Supplementary Fig. 1). To mimic coinfection, the mice were infected with IAV and subsequently infected with SARS-CoV-2. While the terms 'coinfection' and 'superinfection' are often used interchangeably, the use of 'coinfection' here refers to a sequential infection with two viruses within a very short time, with the second infection occurring prior to the elimination of the first virus.

First, the outcome of SARS-CoV-2 infection was tested two dpIi during the presymptomatic stage of influenza (Supplementary Fig. 1a). At this stage, mice do not present any disease manifestations, but the viral titer of IAV in the lungs is high[16]. The IAV-infected mice started losing weight at 5 dpIi and exhibited maximal morbidity at 9–10 dpIi (75% of their initial body weight) (Fig. 1a). At 11 dpIi, all the mice began to gain weight and returned to their initial body weight by 18 dpIi. Remarkably, the mice infected with SARS-CoV-2 at two dpIi exhibited earlier and increased weight loss compared to the mice with IAV infection alone. Moreover, in contrast to the survival and only 38% mortality of the only IAV- and only SARS-CoV-2-infected mice, respectively, all coinfected mice died by 5–7 days post SARS-CoV-2 infection (dpSi) (Fig. 1b, ****$p < 0.0001$).

Subsequently, we examined coinfection with SARS-CoV-2 at five dpIi during the early symptomatic stage. The IAV-infected and coinfected mice started to lose weight at 6-7 dpIi (Fig. 1c). However, while the IAV-infected mice reached their maximal weight loss at 8 dpIi (79% of their initial weight), the coinfected mice continued to lose weight until 10 dpIi (73% of their initial weight). Additionally, the recovery period of the coinfected mice was prolonged compared to that of the IAV-infected mice. While the IAV-infected mice reached 97% of their initial weight at 11 dpIi, the coinfected mice reached 94% of their initial weight by 17 dpIi and returned to their baseline weight only 22 dpIi (Fig. 1c). Moreover, coinfection with SARS-CoV-2 at 5 dpIi resulted in 70% mortality compared to the 43% mortality observed among the mice infected with SARS-CoV-2 alone ($p = 0.08$) (Fig. 1d). Finally, we assessed coinfection when SARS-CoV-2 was administered in the late-symptomatic stage of influenza disease during which maximal morbidity was detected (8 dpIi, Fig. 1e, f). Interestingly, SARS-CoV-2 infection at 8 dpIi had no effect on the body weight or survival rate of the mice. These results suggest that infection with SARS-CoV-2 early upon infection with IAV results in a more severe disease, while a later SARS-CoV-2 infection has no severe effect.

In the human population, coinfection is most likely to occur during the asymptomatic period. Additionally, during the asymptomatic stage of influenza disease, as represented by the experiments with SARS-CoV-2 infection at 2 dpIi (Fig. 1a, b), coinfection resulted in the most severe and lethal disease. Therefore, we chose to focus on this stage, and hereafter, coinfection refers to infection with IAV followed by infection with SARS-CoV-2 two days later.

**Increased IAV and decreased SARS-CoV-2 viral loads in coinfected mice**. To correlate the severe outcome observed in the coinfected mice and the viral load, the IAV and SARS-CoV-2 viral loads were quantified in the lungs, nasal turbinates (N.T.), and brains (Fig. 2). A significant increase in the IAV RNA levels was observed in the lungs of the coinfected mice at 4 dpIi, but not at 6 dpIi, compared to those in the mice with IAV infection (4.6-fold increase). In the N.T., a significant increase in the IAV viral titer was observed at both 4 and 6 dpIi (11-fold and 2.3-fold increases, respectively) in the coinfected mice compared to that in IAV-infected mice (Fig. 2b, c), which coincided with the exacerbated disease. In contrast, the SARS-CoV-2 load was reduced in the coinfected mice compared to that in the SARS-CoV-2-infected mice in both the lungs and the N.T. (albeit nonsignificantly) at 4 and 6 dpIi (Fig. 2d, e). These findings are consistent with previous evidence of pathogenic competition between respiratory viruses, such as influenza and seasonal coronaviruses[17,18]. A possible explanation for the reduced SARS-CoV-2 viral load might be the induction of the innate immune response activated by the IAV

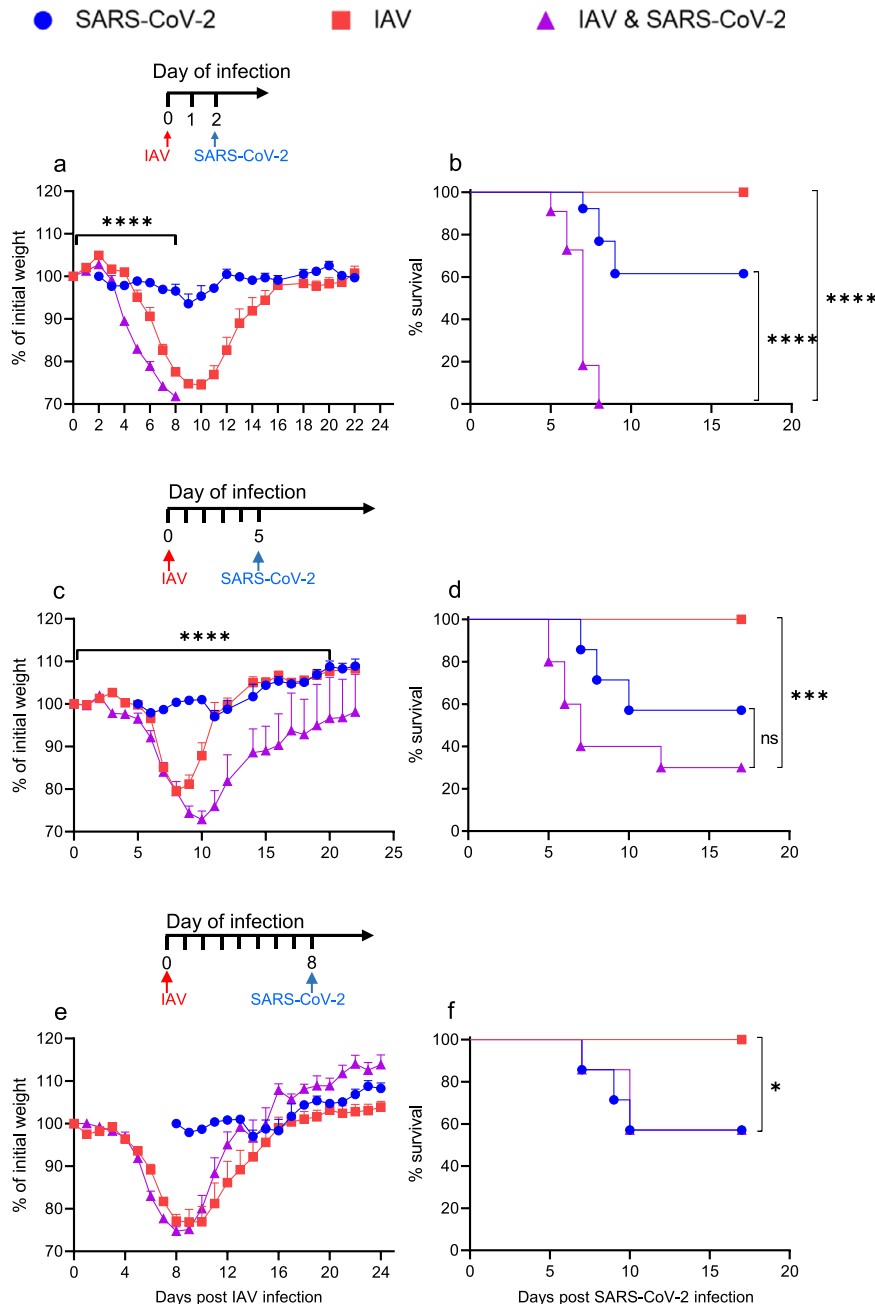

**Fig. 1 Morbidity and mortality in K18-hACE2 mice infected with SARS-CoV-2 following influenza infection.** K18-hACE2 mice were infected with IAV (80 PFU/mouse, i.n.), followed by infection with SARS-CoV-2 (10 PFU/mouse, i.n.) at 2 (**a**, **b**), 5 (**c**, **d**), or 8 (**e**, **f**) dpli. The percent weight loss following infection is shown (**a**, **c**, **e**). The red arrow represents IAV infection. The blue arrow represents SARS-CoV-2 infection. All measurement data are expressed as mean + standard error (SE). The area under the curve (AUC) was calculated using GraphPad Prism 8.4.3; IAV infection compared to coinfection (**a**, **c**, **e**); ****$p < 0.0001$ (**a**, **c**), ns (**e**). Survival curves are shown (**b**, **d**, **f**): *$p = 0.0453$; ***$p = 0.0007$; ****$p < 0.0001$; IAV or SARS-CoV-2 infection compared to coinfection. ns, not significant ($p = 0.0779$), log-rank test (Mantel-Cox). The figure shows one representative experiment of 4 (**a**, **b**), 2 (**c**, **d**), and 1 (**e**, **f**) experiments performed. The figure includes the IAV-infected group, which consisted of 10 (**a**, **b**), 11 (**c**, **d**), or 8 (**e**, **f**) mice; the SARS-CoV-2-infected group, which consisted of 13 (**a**, **b**) or 14 (**c**–**f**) mice; and the coinfection group, which consisted of 11 (**a**, **b**), 10 (**c**, **d**), or 7 (**e**, **f**) mice.

infection prior to the SARS-CoV-2 infection, inhibiting the establishment of infection and replication[19–21]. In addition to innate immunity mechanisms underlying the reduced SARS-CoV-2 viral load, IAV infection may interfere with SARS-CoV-2 infection in vivo. Our data contradict those of a recent study showing that coinfection with IAV enhances SARS-CoV-2 infectivity[22]. In this previous study, no significant differences were found in disease manifestations between coinfected K18-hACE2 mice and SARS-CoV-2-

monoinfected mice[22]. However, the SARS-CoV-2 infection dose in our study is 4-log lower than that in the Bai et al.'s study, and no data comparing coinfected mice and IAV-monoinfected mice were presented by Bai et al., making comparison between the two studies difficult.

Notably, although the SARS-CoV-2 viral load was reduced in the coinfected mice, the remaining levels were sufficient to trigger the lethal outcome of the coinfection.

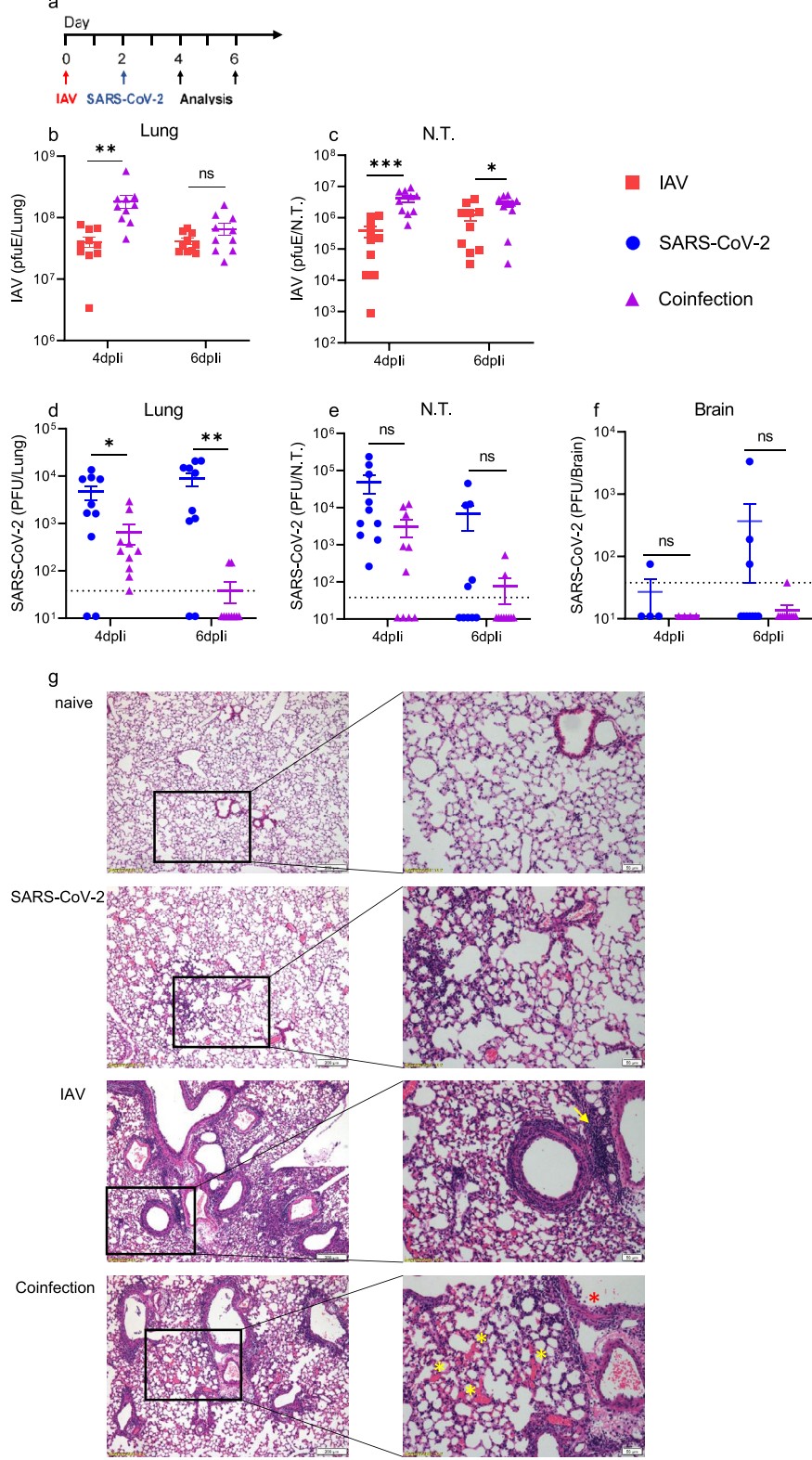

**Histopathological analysis of coinfected K18-hACE2 mice**. Lung pathology was observed in all infected groups and was consistent with the IAV viral load, with the coinfected mice presenting the most severe pathology (Fig. 2g); the lungs of the IAV-infected mice appeared moderately to strongly affected, showing necrosis of the bronchiolar epithelium with massive peribronchiolar infiltration of neutrophils and lymphocytes (Fig. 2g). The SARS-CoV-2-infected lungs showed milder histological changes characterized by minor-to-mild interstitial infiltration of lymphocytes and some neutrophils, with no obvious necrosis (Fig. 2g). The lungs of the coinfected mice showed a histopathological pattern similar to that observed in the IAV group but were more severe, including involvement of the lung parenchyma. Small hemorrhages were observed in the interstitium of some samples, and thickening of the pleura was identified in 3 of 5 animals (Fig. 2g). Taken together, these results

**Fig. 2 Increased IAV and decreased SARS-CoV-2 loads in coinfected mice.** Schematic timeline (**a**). K18-hACE2 mice were sacrificed at 4 or 6 dpli and 2 or 4 dpSi. The IAV RNA load was measured in the lungs (**b**) and nasal turbinates (N.T.) (**c**), and the PFU equivalent per organ (pfuE/organ) was calculated. The SARS-CoV-2 loads in the lungs (**d**), N.T. (**e**), and brains (**f**) were determined by a PFU assay. Each symbol represents one mouse (10 mice per group, except 4 mice for brain testing at 4 dpli). The lines represent the means. The error bars represent the SEs. **$p = 0.0062$ (**b**); ***$p = 0.0008$, *$p = 0.0411$ (**c**); *$p = 0.0212$, **$p = 0.0056$ (**d**); ns, not significant; two-tailed unpaired Student's $t$-test. Hematoxylin and eosin (H&E) staining (**g**) of lung sections from naïve K18-hACE2 mice or after i.n. administration of SARS-CoV-2 (4 dpSi), IAV (6 dpli), or IAV and SARS-CoV-2. Left panel scale bar = 200 µm; right panel is a magnification of the rectangle in the left panel, scale bar = 50 µm. The yellow arrow indicates the affected bronchus surrounded by lymphocytes, with a marked proliferation of the bronchial lymph nodulus. The red asterisk (*) indicates damage to the bronchial epithelium, and the yellow asterisks indicate multiple hemorrhages. Each image is representative of a group of 5 mice.

suggest that the increased morbidity and mortality detected in the coinfected mice are associated with elevated levels of IAV and exacerbated pathology in the respiratory system.

K18-hACE2 mice are a valuable model of SARS-CoV-2 infection and have been studied extensively[23,24]. However, in K18-hACE2 mice, SARS-CoV-2 also enters the brain, which might affect the disease severity[15]. To examine whether the increased morbidity was brain related, SARS-CoV-2 viral load was measured in brain tissues, and was found in only 3 of 10 SARS-CoV-2-infected mice (Fig. 2f; at 4 dpSi). Importantly, no virus was detected in the brains of the coinfected mice, and no brain pathology was apparent at 4 dpSi (Supplementary Fig. 2).

**Coinfection in AdV-hACE2 mice model.** Subsequently, to exclude the possibility that the severe outcome of the coinfected mice is model specific and brain related, we examined an additional independent COVID-19 mouse model, i.e., hACE2-expressing human Ad5 (AdV-hACE2) mice. In the AdV-hACE2 model, C57BL/6J mice were inoculated via the intranasal (i.n.) route with replication-defective adenovirus encoding the hACE2 receptor, resulting in lung-specific pathology[25]. Similar to the K18-hACE2 mice, the AdV-hACE2 mice were significantly more sensitive to coinfection with SARS-CoV-2 at 2 dpIi (Supplementary Fig. 3). While all monoinfected mice survived the infection, only 9% of the coinfected mice survived (Supplementary Fig. 3). These results support that coinfection lethality is not model specific.

**Upregulation of inflammation-related genes in the lungs of coinfected mice.** To elaborate upon the host response to each monoinfection versus coinfection, we assessed the expression of immune-related genes in the lungs at 4 dpIi and 2 dpSi.

The genes tested can be divided into the following groups: the complement system (*C1ra*, *C1rb*, *C3*, and *C1s1*); antigen presentation (*H2-eb1* and *H2-k1*); recruitment and activation of immune cells (*Cd2*, *Fas*, *Cd38*, *Gzma*, and *Prf1*); interleukins (Ils) (*Il-1a* and *Il6*); chemokines (*Ccl5*, *Ccl8*, *Ccl12*, and *Cxcl10*); interferon response (*Ifi44*, *Ifit1*, *Ifit3*, *Ifi27l2a*, and *Irf7*); matrix metalloproteinases (*Mmps*); tissue damage (*Mmp3*, *Mmp8*, *Mmp9*, and *Timp1*); and a member of the Schlafen (*Slfn*) family (*Slfn4*), the ubiquitin-like modifier *Isg15*, and *Z-DNA-binding protein 1* (*Zbp1*).

Overall, in the lungs of the SARS-CoV-2-infected mice, no alterations in the mRNA levels of the tested genes were observed compared to those in the lungs of the uninfected mice (Fig. 3) most likely due to the low infection dose (10 PFU/mouse) and short time post infection (2 days). After the infection with IAV, all tested genes were overexpressed (Fig. 3). Remarkably, the IAV and SARS-CoV-2 coinfection resulted in a significantly larger elevation of gene expression than exhibited upon IAV infection alone, indicating robust induction of the immune system that may have led to exacerbated disease.

**Preexisting immunity to SARS-CoV-2 does not rescue coinfected mice.** Then, we examined whether preexisting immunity to SARS-CoV-2 prevents the severe manifestations in the coinfected mice. Efficient immunity to SARS-CoV-2 was achieved by intramuscular (i.m.) immunization of SARS-CoV-2 (Supplementary Fig. 4a, b). This route of immunization induced both humoral and cellular responses (Supplementary Fig. 4a, b, respectively) against SARS-CoV-2 and was sufficient to protect the mice from a SARS-CoV-2 challenge (Fig. 4a–c). However, while preexisting immunity to SARS-CoV-2 completely prevented the mortality caused by SARS-CoV-2 infection, accounting for 50%, it had no effect on the morbidity and full mortality caused by the IAV and SARS-CoV-2 coinfection (Fig. 4b, c). To rule out the possibility that the inability of SARS-CoV-2 immunization to rescue the coinfected mice was a result of the i.m. immunization route, we examined another SARS-CoV-2 route of immunization. The acquisition of preexisting immunity to SARS-CoV-2 by the i.n. route (Supplementary Fig. 4c, d) did not protect the mice from coinfection (Supplementary Fig. 5), which is similar to the results of infection via the i.m. route, even though both routes of infection induced humoral and cellular responses to SARS-CoV-2 (Supplementary Fig. 4).

**Prior immunity to IAV protected coinfected K18-hACE2 mice.** To determine whether preexisting immunity to IAV can prevent coinfection manifestations, mice were vaccinated i.m. with IAV 30 days prior to the viral infection with IAV and/or SARS-CoV-2. This preexposure to IAV induced both humoral and cellular responses (Supplementary Fig. 6a, b, respectively) against IAV and alleviated the morbidity observed upon infection with IAV (Fig. 4f). Preexposure to IAV had no effect on the survival rate upon SARS-CoV-2 infection (Fig. 4e, $p = 0.64$). Remarkably, immunity to IAV prevented the severe manifestations and mortality caused by the IAV and SARS-CoV-2 coinfection. Neither weight loss nor increased mortality was detected in the coinfected mice vaccinated against IAV compared to those in the coinfected mice without preexisting immunity (Fig. 4e, f).

These data further support the notion that the severe manifestations of coinfection are not the result of a more severe SARS-CoV-2 disease but rather of IAV.

**Prior immunity to IAV rescues coinfected mice and is antibody dependent.** Next, we examined whether the protection of coinfected mice by prior immunization is due to cellular or humoral immunity. First, T-cell depletion was conducted (Supplementary Fig. 7). Mice were immunized by the i.m. route to IAV to induce immunity; after 30 days, CD4$^+$ or CD8$^+$ T cells were depleted, the mice were coinfected with IAV and SARS-CoV-2, and the ability of preexisting immunity to rescue the mice in the presence or absence of T cells was tested (Fig. 4g, h). The depletion of CD4$^+$ or CD8$^+$ T cells had no effect on the survival rate of the preimmune coinfected mice. This result suggests that the protection provided by preimmunity to IAV is T cell independent. Next, to delineate the role of anti-IAV

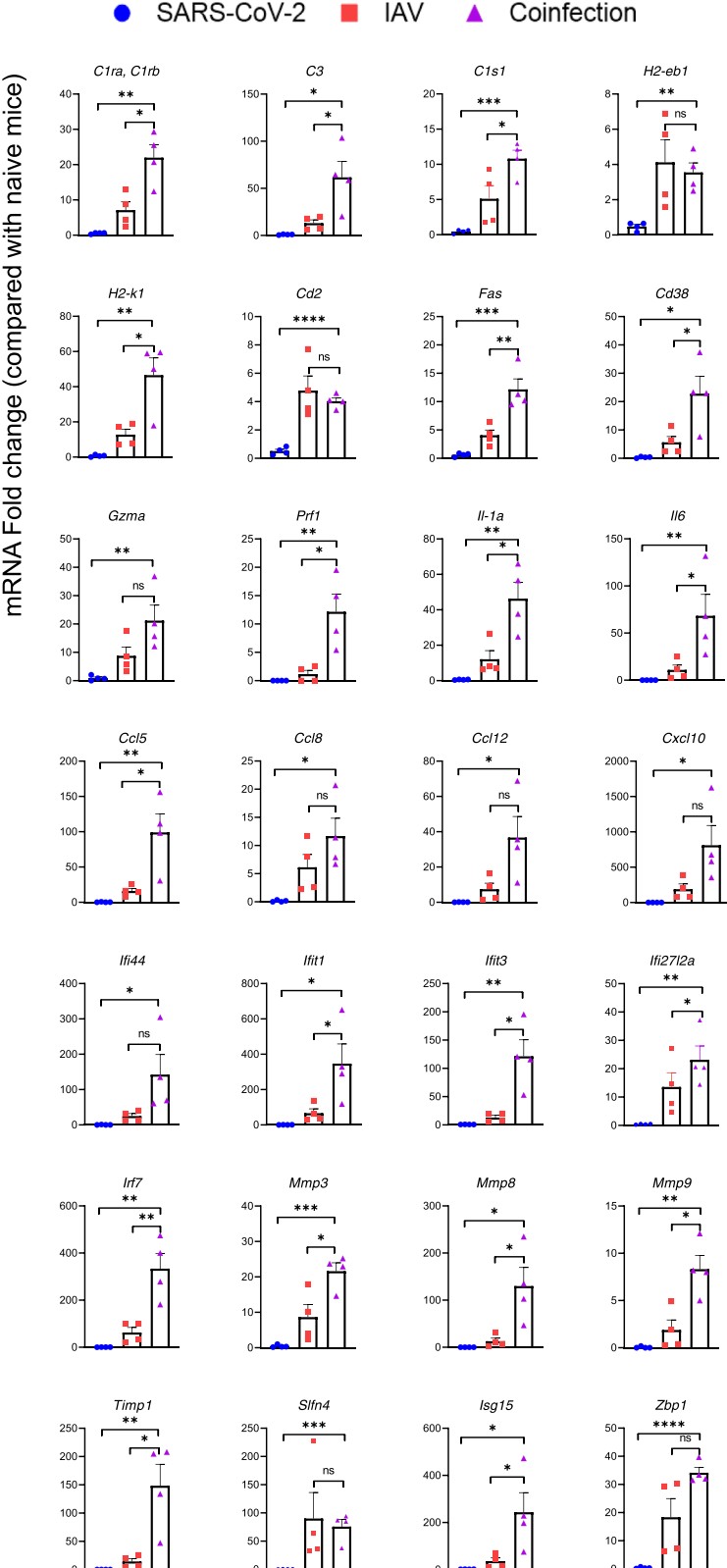

**Fig. 3 Panel of inflammation-related genes with increased expression in the lungs of coinfected mice.** Expression of various inflammation-related genes in the lungs. RNA was isolated from the lungs of K18-hACE2 mice at 4 dpli/2 dpSi and analyzed by real-time qRT-PCR. Each symbol represents one mouse (4 per group). The $Y$ axis represents the fold change in the infected mice compared with the naive mice. The column height represents the mean. The error bars represent the SE. $^*p < 0.05$; $^{**}p < 0.01$; $^{***}p < 0.001$; $^{****}p < 0.00002$  Student's $t$-test two tailed, Microsoft Excel 2016.

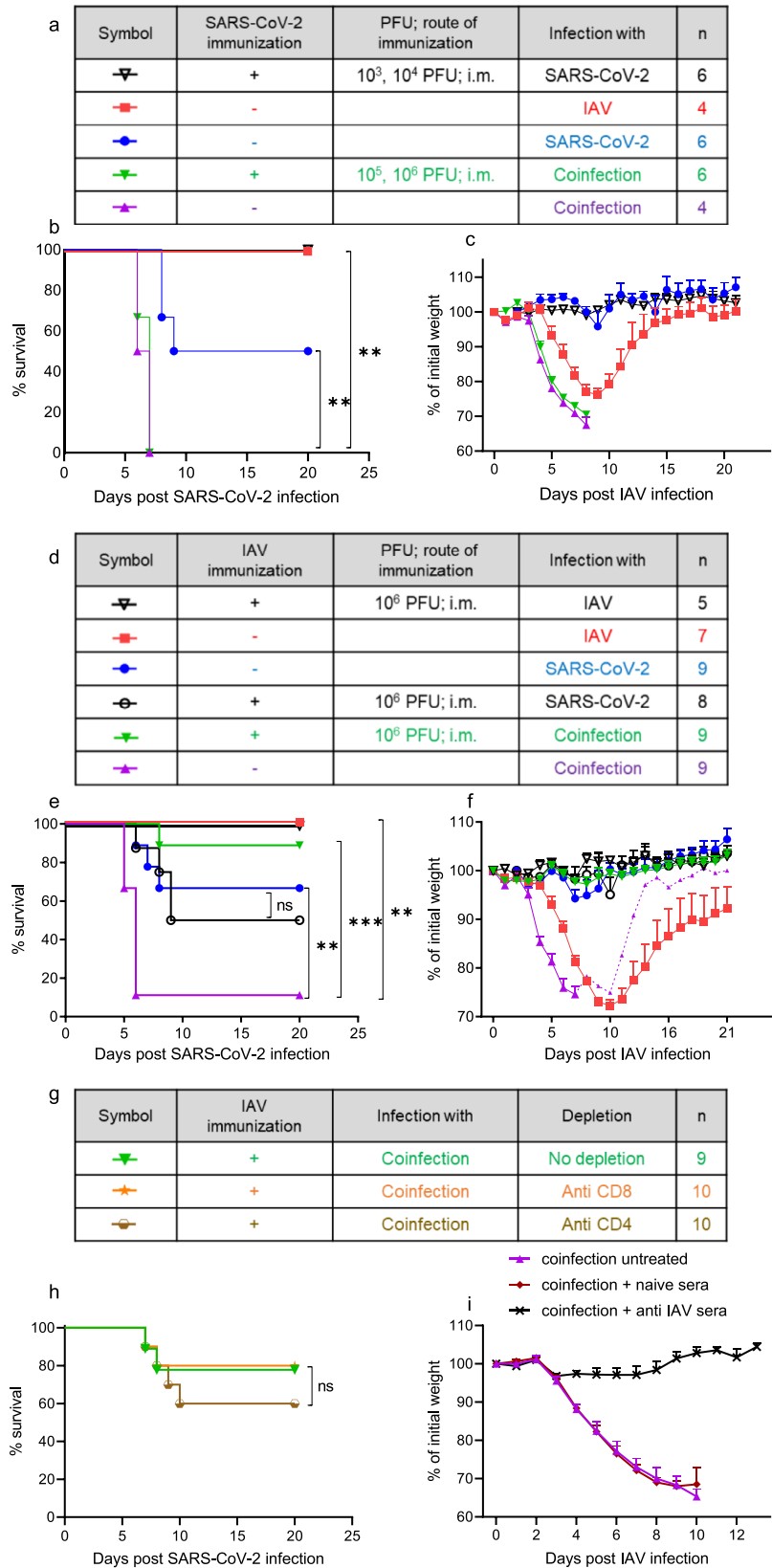

antibodies in the protection of coinfected mice, we performed passive transfer of anti-IAV sera to coinfected mice. Anti-IAV antibodies rescued the coinfected mice (Fig. 4i), indicating that the protection was indeed antibody dependent.

Altogether, our data show that IAV-SARS-CoV-2 coinfection results in severe and lethal disease in mice. The severe manifestations were associated with an elevated IAV viral load, robust induction of innate immunity, and exacerbated pathology in respiratory organs. Importantly, prior humoral immunity to influenza, but not to SARS-CoV-2, prevented disease and mortality, suggesting that influenza intervention by prior vaccination may prove valuable in reducing the risk of severe influenza/SARS-CoV-2 comorbidity.

**Fig. 4 Prior immunity to IAV but not to SARS-CoV-2 protected coinfected K18-hACE2 mice. a** Legend of (**b**, **c**). **d** legend of (**e**, **f**). **g** legend of (**h**). K18-hACE2 mice were immunized by the i.m. route to SARS-CoV-2 or IAV as indicated in (**a**, **d**, **g**). Thirty days post immunization, immunized and nonimmunized mice were infected i.n. with the indicated virus (**b**, **c**, **e**, **f**). Survival curves (**b**, **e**): $^{**}p = 0.0012$ (**b**, SARS-CoV-2 immunization and infection compared to coinfection with no prior immunization); $^{**}p = 0.0018$ (**b**, SARS-CoV-2 infection compared to coinfection); $^{**}p = 0.0039$ (**e**, SARS-CoV-2 infection compared to coinfection); $^{***}p = 0.0005$ (**e**, IAV immunization and infection compared to coinfection); $^{**}p = 0.0019$ (**e**, IAV infection compared to coinfection); ns ($p = 0.6430$), not significant; log-rank test (Mantel-Cox). Percent weight loss following infection (**c**, **f**). All measurement data are expressed as mean + SE. $n$, number of mice per group. The dashed line (**f**) represent 1 surviving mice of 9. **h** Survival curve of IAV-immunized mice coinfected with IAV and SARS-CoV-2 following T cell depletion with anti-CD8 or anti-CD4 antibodies. Anti-CD8 or anti-CD4 antibodies were administered i.p. 1 day prior to viral infection and every 2–3 days during the experiment. Log-rank test (Mantel-Cox), ns, not significant. (**i**) Anti-IAV sera rescued K18-hACE2 coinfected mice. Percent weight loss following infection. Mice were coinfected with IAV and SARS-CoV-2 and treated with anti-IAV sera ($n = 10$), or naive sera ($n = 8$), or left untreated ($n = 7$). Sera were administered i.p. 1 day prior to IAV infection. Error bars represent the standard error (SE).

## Methods

**Cells**. Vero E6 cells (ATCC® CRL-1586TM) and Madin-Darby canine kidney (MDCK) cells (ATCC® CCL-34™) were maintained in Dulbecco's modified Eagle's medium (DMEM) supplemented with 10% fetal bovine serum (FBS), MEM non-essential amino acids, 2 nM L-glutamine, 100 units/ml penicillin, 0.1 mg/ml streptomycin and 12.5 units/ml nystatin (Biological Industries, Israel). The cells were cultured at 37 °C in a 5% $CO_2$ and 95% air atmosphere.

**Viruses**. The SARS-CoV-2 isolate Human 2019-nCoV ex China strain BavPat1/2020 was kindly provided by Prof. Dr. Christian Drosten (Charité, Berlin) through the European Virus Archive – Global (EVAg Ref-SKU: 026 V-03883). The viral stocks were propagated (4 passages) and titrated in Vero E6 cells. The genome sequence has been deposited at the GISAID EpiCoV coronavirus SARS-CoV-2 platform database under the identifier EPI_ISL_3838266.

IAV strain A/Puerto Rico/8/1934 (H1N1) (PR8) was propagated by injecting 0.1 ml of a 1:100 diluted viral stock (2048 hemagglutination units, HAUs) into the allantoic sac of 11-day-old embryonated chicken eggs. After incubation for 72 h at 37 °C, ~9 ml of virus-rich allantoic fluid was collected. The HAU value was determined using chicken erythrocytes, and virus titers were determined by a plaque assay using MDCK cell monolayers[26].

Ad-GFP-hACE2 (code ADV-200183) was purchased from Vector Biosystems Inc. (Malvern, PA, USA).

All viruses were aliquot and stored at −80 °C until use.

**Animal experiments**. All animal experiments involving SARS-CoV-2 were conducted in a biosafety level 3 (BSL3) facility. The animal handling was performed in accordance with the regulations outlined in the U.S. Department of Agriculture (USDA) Animal Welfare Act and the conditions specified in the Guide for Care and Use of Laboratory Animals (National Institutes of Health, 2011). The animal studies were approved by the ethical committee for animal experiments of the Israel Institute for Biological Research (IIBR) (protocol numbers M-29-20, M-39-20, M-40-20, M-41-20, M-36-21, and M-37-21). Female K18-hACE2 transgenic mice (B6. Cg-Tg(K18-ACE2)2Prlmn/J; #034860) and female and male C57BL/6J mice (Jackson Laboratory) (6–8 weeks old) were maintained at 20–22 °C and a relative humidity of 50 ± 10% under a 12-hour light/dark cycle. The animals were fed commercial rodent chow (Koffolk Inc.) and provided tap water ad libitum. Prior to infection, the mice were kept in groups of 10. The mice were randomly assigned to the experimental groups.

For the PR8 and SARS-CoV-2 infection, the viruses were diluted in phosphate-buffered saline (PBS) supplemented with 2% FBS (Biological Industries, Israel). Anesthetized animals (ketamine 75 mg/kg and xylazine 7.5 mg/kg in PBS) were infected by i.n. instillation of 20 µl (PR8: 80 PFU/mouse, SARS-CoV-2: 10 PFU/mouse).

For the infection with Ad-GFP-hACE2[25], C57BL/6J mice were injected intraperitoneally (i.p.) with 2 mg of an anti-IFNAR (interferon-α/β receptor) monoclonal antibody (mAb) (LEINCO Technologies, Inc., St. Louis, MO, USA, code I-401-100) in 0.5 ml. After 24 h, 20 µl of Ad-GFP-hACE2 were administered to the anesthetized mice. After 3 days, the mice were treated i.n. with 80 PFU of PR8, and after 2 days, the mice were treated i.n. with $10^5$ PFU of SARS-CoV-2 (all i.n. administrations were performed under anesthesia).

*Animal immunization*. For the SARS-CoV-2 immunization, i.n. instillation of 2 PFU/mouse or i.m. injection of $10^3$, $10^4$, $10^5$, or $10^6$ PFU/mouse SARS-CoV-2 was performed. For the IAV immunization, the mice were vaccinated i.m. with $10^6$ PFU/mouse. The immunized mice were infected 30 days post immunization.

**Determination of the viral load in organs**. The viral loads were determined at 2 and 4 dpSi or 4 and 6 dpIi. In the coinfected group, the viral load was determined at 4 and 6 dpIi (2 and 4 dpSi). Each group included 10 mice, except for the experiments examining brains at 2 dpSi, which included a group of 4 K18-hACE2 animals. The lungs, N.T., and brains were harvested and stored at −80 °C until further processing. The organs were processed for titration in 1.5 ml of ice-cold PBS. Tissues were homogenized (ULTRA-TURAX® IKA R104) for 30 s in 1.5 ml of ice-cold PBS, followed centrifugation (270g, 10 min, 4 °C) and collection of the supernatants for virus titration. Processed tissue

homogenates were divided for immediate RNA extraction for the IAV RNA determination and gene expression analysis or kept at −80 °C until further processing for the viral titration (used for SARS-CoV-2 PFU assay).

The SARS-CoV-2 viral load was determined using a PFU assay[27]. Serial dilutions of extracted organ homogenates from mice infected with SARS-CoV-2 or coinfected with IAV and SARS-CoV-2 were prepared in infection medium (MEM containing 2% FBS) and used to infect Vero E6 monolayers in duplicate (200 µl/well). The plates were incubated for 1 h at 37 °C to allow viral adsorption. Then, 2 ml overlay (MEM containing 2% FBS and 0.4% tragacanth; Merck, Israel) were added to each well, and the plates were incubated at 37 °C in a 5% $CO_2$ atmosphere for 48 h. The medium was then aspirated, and the cells were fixed and stained with 1 ml/well crystal violet solution (Biological Industries, Israel). The number of plaques per well was determined, and the SARS-CoV-2 PFU titer was calculated.

The IAV RNA level was determined using real-time RT-PCR (see below).

**Quantitative real-time RT-PCR**. RNA was extracted with a viral RNA mini kit (Qiagen, Germany) according to the manufacturer's instructions. The IAV RNA loads in the lung and N.T. were determined by quantitative RT-PCR (qRT-PCR). Real-time RT-PCR was conducted with a SensiFAST™ Probe Lo-ROX One-Step Kit (Bioline, 78005) and analyzed with a 7500 Real-Time PCR System (Applied Biosystems). The PFU equivalent per organ (pfuE/organ) was calculated from a standard curve generated from virus stocks. The quantitative PCR (qPCR) primers and probes used for the detection of PR8 were PR8-PA-FW: CGGTCCAAATTCCTGCTGA; PR8-PA-RW: CATTGGGTTCCTTCCATCCA; and PR8-PA-Probe: CCAAGTCATGAAGGAGAGGGAATACCGCT.

Total RNA extracted from the lungs of mice infected with IAV or SARS-CoV-2 or coinfected at 2 dpSi and 4 dpIi was used to measure the differential expression of genes by qRT-PCR using corresponding specific primers printed on 96-well plates (Custom TaqMan Array Plates, Applied Biosystems™) as previously described[28]. Briefly, 1 µg of cDNA was synthesized from RNA using a Verso cDNA Synthesis Kit (Thermo Fisher Scientific, Waltham, MA, USA) according to the manufacturer's instructions. The samples were subjected to qPCR with TaqMan® Fast Advanced Master Mix (7500 Real-Time PCR System, Applied Biosystems, Thermo Fisher Scientific). The housekeeping gene GAPDH was used to normalize the fold change of each gene compared to the mock-infected control at the same time point, which was calculated as ΔΔCT.

**Histopathology**. For the hematoxylin and eosin (H&E) general histopathology evaluation, 6 dpIi (4 dpSi) mice ($n = 5$ per group) were anesthetized and then perfused transcardially with PBS, followed by 4% paraformaldehyde. The lungs and brains were isolated and fixed in 4% paraformaldehyde at room temperature for 2 weeks. The fixed tissues were sent to 'Patho-Logica' (Rehovot, Israel) for processing and analysis. Serial coronal 4 µm thick sections were obtained, and selected sections were stained with H&E for light microscopy examination. Pictures were taken using an Olympus microscope (BX60, serial No. 7D04032) equipped with a microscope camera (Olympus DP73, serial No. OH05504) at objective magnifications of X4 and X10.

**Plaque reduction neutralization test**. The SARS-CoV-2-neutralizing antibody levels were determined using a plaque reduction neutralization test (PRNT)[27]. Briefly, all sera were heat inactivated at 60 °C for 30 min, then serially diluted twofold in 400 µl of infection medium, mixed with 400 µl of 300 PFU/ml SARS-CoV-2, and incubated at 37 °C in an atmosphere of 5% $CO_2$ for 1 h. Then, 200 µl of each serum/virus mixture was added in duplicate to Vero E6 cell monolayers, and the cells were incubated for 1 h at 37 °C. A virus mixture without serum was used as a control. Two milliliters overlay were added to each well, and the plates were incubated at 37 °C in a 5% $CO_2$ atmosphere for 48 h. The medium was then aspirated, and the cells were fixed and stained with 1 ml/well crystal violet solution. The number of plaques per well was determined, and the serum dilution that neutralized 50% of the virions (NT50) was calculated using Prism software (GraphPad Software Inc.).

**ELISpot assays**. For the detection of IFNγ-secreting cells from mice immunized with SARS-CoV-2 or IAV, the mice were sacrificed 7 or 25 days post immunization, respectively. The spleens were dissociated in GentleMACS C-tubes (Miltenyi Biotec),

filtered, separated on Ficoll-Paque (GE), and washed with medium. Then, $4 \times 10^5$ cells from each sample were plated into 96-well ELISpot plates in duplicate in the presence of peptides representing H-2Kb immunodominant epitopes at a final concentration of 2 µg/ml and incubated at 37 °C for 24 h. The naive samples consisted of pools from 2 mice. The frequency of IFNγ-secreting cells was determined using a Mouse IFN-γ Single-Color ELISpot kit (Cellular Technology Limited, Germany) according to the manufacturer's instructions. The frequency of cytokine-secreting cells was quantified with an ImmunoSpot S6 Ultimate reader and analyzed with ImmunoSpot software (Cellular Technology Limited, Germany). The following Db-restricted antigens were used for the stimulation:

influenza PR8-derived peptide: NP366–374 ASNENMETM
SARS-CoV-2-derived peptide: Spike 539-546 VNFNFNGL

For all assays, antigen-free cells supplemented with medium were used as a negative control.

**Enzyme-linked immunosorbent assay**. Nunc MaxiSorp enzyme-linked immuno-sorbent assay (ELISA) plates (Thermo Fisher Scientific, USA) were coated with PR8 virus (whole virus, sucrose purified) in carbonate/bicarbonate solution (Sigma, Israel Cat. C3041) at 4 °C overnight. Plates were blocked with TSTA buffer (50 mM Tris, pH 7.6 + 140 mM NaCl + 0.05% Tween 20 + 2% BSA) for 60 min at 37 °C. Following blocking and washing with PBST buffer (PBS + 0.05% Tween 20), the plates were incubated with a naive or PR8 preimmune mouse serum diluted from 1:400 to 1:52,800 for 1 h at 37 °C. Following washing, alkaline phosphatase-conjugated anti-mouse IgG was used (diluted 1:1000) as a secondary antibody (Sigma, Israel, cat. A5153). P-nitrophenyl phosphate substrate (Sigma, Israel, cat. N2770) was added after washing, and the optical density was measured (SpectraMax 190 microplate reader, Molecular Devices, Sunnyvale, CA, O.D. at 405 nm) after 60 min of incubation at room temperature. The anti-PR8 IgG values were determined by subtracting twice the value of the control (sera from naive mice) from the examined sample.

**In vivo T cell depletion**. Rat anti-mouse CD4 (clone GK1.5, ATCC TIB-207) and CD8 (clone 2.43, ATCC TIB-210) antibodies were produced and purified in our laboratory. Anti-CD4 or anti-CD8 antibodies were administered (200 µg per i.p. injection) to K18-hACE2 mice 1 day (−1 day) before viral infection and every 2–3 days during the first 12 days of the experiment. The CD4 or CD8 depletion was verified by flow cytometry (Supplementary Fig. 7).

**Flow cytometry**. Splenocytes were collected as described for the ELISpot assay. The cells were stained with Aqua Live/Dead Cell Stain (Thermo Fisher, L34966), blocked with an anti-CD16/32 FcγR antibody for 15 min and subsequently stained with fluorescently labeled antibodies for 30 min. The following antibodies from e-Bioscience (Thermo Fisher Scientific) were used: PE anti-CD3ε (clone 145-2C11, diluted 1:200), Alexa Fluor 700 anti-CD4 (clone RM4-5, diluted 1:800), and APC anti-CD8 (clone 56–6.7, diluted 1:800). All washing procedures were performed using flow buffer composed of PBS, 2% FBS and 0.05% NaN₃. The samples were collected using a Fortessa Flow Cytometer (BD Biosciences) and analyzed with FlowJo software (TreeStar).

**Passive immunization**. Six to eight week-old female K18-hACE2 mice were immu-nized against IAV by i.m. injection of PR8 ($10^6$ PFU/mouse). Four weeks later, sera were collected and pooled, and the levels of anti-PR8 IgG antibodies in the anti-IAV sera were determined. The sera were administered (i.p. 200 µl/mouse) 1 day prior to infection with IAV. Mice were coinfected as described above.

**Reporting summary**. Further information on research design is available in the Nature Research Reporting Summary linked to this article.

## Data availability

The data that support the findings of this study are available from the corresponding author upon reasonable request. Source data are provided with this paper.

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

## Acknowledgements

The authors would like to thank Prof. Dr. Christian Drosten at the Charité Uni-versitätsmedizin, Institute of Virology, Berlin, Germany, for providing the SARS-CoV-2 BavPat1/2020 strain. We thank Dr. Michal Mandelboim for IAV/Puerto Rico/8/34 H1N1 (PR8). We thank Dr. Ofer Cohen (IIBR) for help with the AdV-hACE2 model.

## Author contributions

H.A., E.B.V., B.P., S.M., and T.I. designed the research. H.A., E.B.V., B.P., S.M., R.A., H.T., Y.Y.R., N.E., L.C., M.A., Y.V., D.G., E.M., N.P., and T.I. performed the experiments. H.A., E.B.V., N.P., and T.I. wrote the manuscript. S.W., T.C., and E.B.H. contributed in technical support. All authors discussed the results and commented on the manuscript before sub-mission. R.A. is supported by the Israel Science Foundation (grant 521/18). E.B.V. is sup-ported by the Katzir Foundation.

## Competing interests

The authors declare no competing interests.
