## [Peer Review File. · Nature Communications]

Increased lethality in influenza and SARS-CoV-2 coinfection is prevented by influenza immunity but not SARS-CoV-2 immunityEditorial Note: This manuscript has been previously reviewed at another journal that is not operating a transparent peer review scheme. This document only contains reviewer comments and rebuttal letters for versions considered at Nature Communications.

Reviewers' Comments:

Reviewer #1:

Remarks to the Author:

The authors have addressed my concerns. The additional experiments have strengthened the manuscript.

Reviewer #2:

Remarks to the Author:

The paper by Achdout et al. studies the effect of SARS-CoV-2/influenza superinfection in a transgenic mouse model expressing human ACE2 under the K18 promoter. The data obtained in this model is compared with that obtained in C57BL/6 mice in which ACE2 is delivered via Adenovirus infection. The study indicates that superinfection results in severe respiratory disease. This is precluded by previous immunity to flu but not SARS-CoV-2, and this immunity is antibody dependent.

This is a great study, well conducted and with conclusions of obvious public health relevance. I only have some minor suggestions as indicated below. Well done.

1- In the abstract (Line 27), I think antibody-dependent is more adequate than humoral-dependent. Alternatively 'dependent on humoral immunity'.

2- Why do the authors conclude that 'in the human population, coinfection is most likely to occur during the asymptomatic period'? This for sure does not apply for mild cases of COVID-19 which are the vast majority.

3- I think that throughout the paper, superinfection is more appropriate than coinfection

4- Did the authors sequence their SARS-CoV-2 stocks? This is important in view of the furin cleavage deletions appearing as a consequence of passage in Vero's. This should be indicated

5- it would be easier for the reader if the color codes of the graphs were maintained in all the figures

REVIEWERS' COMMENTS	Answer
REVIEWERS' COMMENTS Reviewer #1 (Remarks to the Author): The authors have addressed my concerns. The additional experiments have strengthened the manuscript.	We wish to thank the Reviewer for his valuable remarks. We also believe that the additions made following his remarks strengthened the manuscript.
REVIEWERS' COMMENTS Reviewer #2 (Remarks to the Author): The paper by Achdout et al. studies the effect of SARS-CoV-2/influenza superinfection in a transgenic mouse model expressing human ACE2 under the K18 promoter. The data obtained in this model is compared with that obtained in C57BL/6 mice in which ACE2 is delivery via Adenovirus infection. The study indicates that superinfection results in severe respiratory disease. This is precluded by previous immunity to flu but not SARS-CoV-2, and this immunity is antibody dependent. This is a great study, well conducted and with conclusions of obvious public health relevance. I only have some minor suggestions as indicated below. Well done.	We wish to thank the Reviewer for his comments and remarks. Our replay for his suggestions appears below. The line numbers specified are for the track changes version.
1- In the abstract (Line 27), I think antibody-dependent is more adequate than humoral-dependent. Alternatively 'dependent on humoral immunity'.	As suggested, changed to antibody-dependent.
2- Why do the authors conclude that 'in the human population, coinfection is most likely to occur during the asymptomatic period'? This for sure does not apply for mild cases of COVID-19 which are the vast majority.	We are sorry for the misunderstanding. We added clarification for the meaning of asymptomatic period of influenza (lines 112-113).
3- I think that throughout the paper, superinfection is more appropriate than coinfection	We address the terms coinfection and superinfection in lines 81-84. ("While the terms coinfection and superinfection are often used interchangeably, the use of 'coinfection' here refers to a sequential infection with 2 viruses within a very short time, with the second infection occurring prior to the elimination of the first virus.").

4- Did the authors sequenced their SARS-CoV-2 stocks? This is important in view of the furin cleavage deletions appearing as a consequence of passage in Vero's. This should be indicated	Yes. GISAID submission: EPI_ISL_3838266. No deletion in the furin cleavage site is noted. Data was added to the text, lines 262-264.
5- it would be easier for the reader if the color codes of the graphs were maintained in all the figures	The color codes of the groups are maintain in all figures when possible.